



# Intraseasonal variation of the northeast Asian anomalous anticyclone and its impacts on air pollution in the North China Plain in early winter

Xiadong An[1,2], Wen Chen[2], Peng Hu[2], Shangfeng Chen[2], Lifang Sheng[1,3]

[1]Department of Marine Meteorology, College of Oceanic and Atmospheric Sciences, Ocean University of China, Qingdao 266100, China

[2]Center for Monsoon System Research, Institute of Atmospheric Physics, Chinese Academy of Sciences, Beijing 100190, China

[3]Ocean-Atmosphere Interaction and Climate Laboratory, Key Laboratory of Physical Oceanography, Ocean University of China, Qingdao 266100, China

*Correspondence to*: W. Chen (chenw@mail.iap.ac.cn), L. F. Sheng (shenglf@ouc.edu.cn)

**Abstract.** The canonical view of the northeast Asian anomalous anticyclone (NAAA) is a crucial factor for determining poor air quality in the North China Plain (NCP) on the interannual timescale. However, there is considerable intraseasonal variability in the NAAA in early winter (November to January), and the corresponding mechanism of its impacts on air pollution in the NCP is not well understood. Here, we find that the intraseasonal NAAA usually establishes quickly on day −3 with a life span of eight days, and its evolution is closely tied to the Rossby wave from upstream (i.e., the North Atlantic). Moreover, we find that the NAAA with a westward tilt might be mainly related to the wavenumbers 3−4. Further results reveal that under this background, the probability of regional air pollution for at least three days in the NCP is as high as 69% (80% at least two days) in NDJ period 2000−2021. In particular, air quality in the NCP tends to deteriorate on day 2 prior to the peak day of the NAAA and reaches a peak on day −1 with a life cycle of four days. In the course of air pollution, a shallower atmospheric boundary layer and stronger surface southerly wind anomaly associated with the NAAA in the NCP appear 1 day earlier than poor air quality, which provides dynamic and thermal conditions for the accumulation of pollutants and finally occurrence of the air pollution on the following day. Furthermore, we show that the stagnant air leading to poor air quality is determined by the special structure of temperature in the vertical direction of the NAAA, while weak ventilation conditions might be related to a rapid buildup of the NAAA. The present results quantify the impact of the NAAA on air pollution in the NCP on the intraseasonal timescale.



# 1 Introduction

The North China Plain (NCP, 32–42ºN, 110–120ºE) has undergone a series of air pollution episodes, particularly in late autumn and early winter (Wang et al., 2019; Yin et al., 2021), which is recognized as significant human health risk and economic activity (Geng et al., 2021a). Nevertheless, $PM_{2.5}$ pollution in China has been successfully reduced (i.e., $PM_{2.5}$ concentrations fell by 42% between 2013 and 2018 across 74 large cities in China), thanks to comprehensive emission control in response to mounting public health risks (Cleaner air for China, 2019). However, $PM_{2.5}$ concentration in this

region remains the highest in the world (Jeong et al., 2021). Additionally, air pollution is not only related to emissions (i.e., its long-term trends), but also is modulated by the atmospheric circulation (i.e., short-term seasonal variability) (Yang et al., 2016; Dang and Liao, 2019). Moreover, Cai et al. (2017) found that global warming will further increase the incidence of haze days in China by reducing the wind strength.

Specific to air pollution in the NCP, previous studies have found many influence factors, including sea surface temperature

(Chang et al., 2016; Jeong et al., 2018; Yu et al., 2020), Arctic sea-ice (Wang et al., 2015, Zou et al., 2020), Eurasian snow cover and soil (Zou et al., 2017; Yin et al., 2018) and climate internal variability including the Eurasian Teleconnection (Li et al., 2019) and subtropical westerly jet waveguide (An et al., 2020, 2021; Mei et al., 2021), and so on. As a matter of fact, the northeast Asian anomalous anticyclone (NAAA) is directly related to air pollution in the NCP (Wang et al., 2020; Callahan and Mankin, 2020). Wang et al. (2009) and Song et al. (2016) found a weak East Asian trough is usually related to

the NAAA, which is mainly induced by the low-frequency Rossby wave and synoptic transient eddy. As a synoptic system, the NAAA not only leads to higher temperature over East Asia by weakening East Asian trough (Song et al., 2016), but also directly modulating stagnant and ventilated conditions for air pollution in the NCP (e.g., Chang et al., 2016; Zhong et al., 2019). Moreover, the interannual variability of the NAAA is regulated by external factors mentioned above via atmospheric teleconnection (Yin et al., 2017; Wang et al., 2020; An et al., 2020). Therefore, the NAAA can't be ignored when studying

meteorological causes of air pollution in the NCP.

Although previous studies have demonstrated that the NAAA is the decisive factor affecting interannual variation of wintertime air pollution in the NCP except emissions (An et al., 2020; Wang et al., 2020), the role of the NAAA on air pollution on the intraseasonal timescale requires further investigation. On the synoptic scale, Zhong et al. (2019) found that the NAAA also plays a crucial role in haze of the NCP in December. For the research within the intraseasonal timescale,

however, the existing studies mainly focus on the analysis of some haze cases (i.e., haze cases are limited to December in the years 2014−2016) (Zhong et al., 2019), lacking a more quantitative statistical analysis and further mechanistic analysis. Therefore, this study focuses on influence of the NAAA on air pollution on the intraseasonal timescale. With the objectives as follows: to derive the characteristics of air pollution evolution in the NCP under the background of the NAAA in November to January (NDJ) on the intraseasonal timescale; to assess the probability of the NAAA in relation to air pollution





in the NCP; and to further explore physical mechanisms of the NAAA deriving meteorology conditions for air pollution in the NCP.

The rest of this study is organized as follows: Section 2 describes the data and methods used in this paper. The results of this paper are included in Sect. 3. Specifically, the NAAA events and associated weather patterns are described in Sect. 3.1. Air pollution in the NCP related to the NAAA is described in Sect. 3.2. Section 3.3 and 3.4 introduce the physical mechanisms

of the NAAA causing air pollution. The paper concludes with a brief summary and discussion in Sect. 4.

## 2 Data and methods

### 2.1 Data

The monthly and daily reanalysis data were mainly obtained from the National Center for Environmental Prediction (NCEP)/National Center for Atmospheric Research (NCAR) Reanalysis 2 dataset (Kanamitsu et al., 2002). The dataset

extends from 1979 to present, with a spatial resolution of 2.5° × 2.5° and 17 vertical layers extending from 1000 to 10 hPa. The variables including zonal and meridional wind, and air temperature are daily data. The geopotential height is monthly and daily data. In addition, daily atmospheric boundary layer height (ABLH) with a spatial resolution of 1.0° × 1.0° in this study averaged from the 6-hourly dataset was taken from the fifth generation the European Centre for Medium-Range Weather Forecasts (ECMWF) reanalysis (ERA5, Hersbach et al., 2018).

Air quality degradation is often accompanied by high $PM_{2.5}$ concentration (e.g., Yang et al., 2016; Dang and Liao, 2019). Consequently, $PM_{2.5}$ concentration is used to describe air pollution in this study. The daily $PM_{2.5}$ concentration data used in this study is a near real-time air pollutant database known as Tracking Air Pollution in China (TAP, http://tapdata.org.cn/). The daily TAP $PM_{2.5}$ concentration data extends from 2000 to present, with a spatial resolution of 10 km in China, which combines information from multiple data sources like ground observations, satellite aerosol optical depth, operational

chemical transport model simulations, and other ancillary data (i.e., meteorological fields, land use data, population and elevation) (Geng et al., 2021b). According to Geng et al. (2021b), the TAP $PM_{2.5}$ concentration is estimated based on a two-stage machine learning model coupled with the synthetic minority oversampling technique and a tree-based gap-filling method, which has an averaged out-of-bag cross-validation $R^2$ of 0.83 for different years (Geng et al., 2021b), which is widely used in air pollution research (e.g., Geng et al., 2021a). The results from the TAP $PM_{2.5}$ concentration are generally

consistent with the observed $PM_{2.5}$ concentration data during December 2014 to January 2021, which can be downloaded at website https://quotsoft.net/air/ (not shown).

It is noteworthy that the anomalies of the reanalysis data in this paper were calculated using climatology covering the period 1981–2020. Especially, to remove long-term trend due emission and a comprehensive emission control of Chinese





government (Cleaner air for China, 2019), $PM_{2.5}$ concentration anomaly were calculated based on a 3-year running

climatology state.

## 2.2 Methods

Firstly, we get a spatial pattern of the NAAA using the Empirical Orthogonal Function (EOF) based monthly mean geopotential height anomaly over domain 25º–55ºN, 100º–160ºE in NDJ period 1979−2021 (Figs. 1a and b). The first EOF mode (EOF1) represents the NAAA (Fig. 1a), which explains a total variance of 44.2% and is well separated from the other

eigenvalues as per the criterion of North et al. (1982). To obtain the typical NAAA on the intraseasonal timescale, the NDJ 8–90-day Butterworth bandpass-filtered daily geopotential height anomaly field at 500 hPa in the region 25º–55ºN, 100º–160ºE is projected onto the EOF1 to obtain a daily principal component (PC, hereafter) time series. Specifically, z is defined as the observed daily geopotential height anomaly field at 500 hPa, which is projected onto the EOF1 spatial pattern (e) to obtain the PC time series (Fig. 1c) (Baldwin et al. 2009):

$PC = \frac{ze}{e^T e}.$                                                                                  (1)

Second, the typical NAAA events is defined as the following way (Fig. 2). First, we rank the values of PC time series in descending order to select the date with the largest PC (i.e., the peak day). If the PC values on at least three days centered on the peak day all exceed one standard deviation, then this peak day is marked as day 0 of a strong NAAA event. Once a day 0 is found, no day within twenty-one days of the central date (day 0) can be defined as a strong NAAA event. This procedure

prevents the algorithm from counting the same strong NAAA event repeatedly. Third, we repeat the above procedure until the values of PC don't exceed one standard deviation to guarantee that all the strong NAAA events are identified. Based on the above criterion (Fig. 2), 94 NAAA events in NDJ period 1979−2021 are selected in this study. This method is similar to that of Franzke et al. (2011), who studied the Pacific–North American teleconnection. In addition, the same method was used by Song et al. (2016), who studied the intraseasonal variation of the East Asian trough in winter.

In addition, to examine the propagation of anomalous Rossby waves generating the NAAA, we calculated the horizontal stationary wave activity flux (WAF), as defined by Takaya and Nakamura (2001). Daily reanalysis data; i.e., the zonal wind, meridional wind, and anomalous geopotential height, are used to calculate the vector **W**.

$$\boldsymbol{W} = \frac{p\cos\varphi}{2|\boldsymbol{U}|} \cdot \begin{pmatrix} \frac{U}{a^2\cos^2\phi}\left[\left(\frac{\partial\psi'}{\partial\lambda}\right)^2 - \psi'\frac{\partial^2\psi'}{\partial\lambda^2}\right] + \frac{V}{a^2\cos\phi}\left[\frac{\partial\psi'}{\partial\lambda}\frac{\partial\psi'}{\partial\phi} - \psi'\frac{\partial^2\psi'}{\partial\lambda\partial\phi}\right] \\ \frac{U}{a^2\cos\phi}\left[\frac{\partial\psi'}{\partial\lambda}\frac{\partial\psi'}{\partial\phi} - \psi'\frac{\partial^2\psi'}{\partial\lambda\partial\phi}\right] + \frac{V}{a^2}\left[\left(\frac{\partial\psi'}{\partial\phi}\right)^2 - \psi'\frac{\partial^2\psi'}{\partial\phi^2}\right] \end{pmatrix},$$                (2)

where $\boldsymbol{W}$ is the wave activity flux (unit: $m^2\ s^2$), $\psi$ $(= \Phi/f)$ is the geostrophic stream function, $\Phi$ (unit: m) is geopotential

height, and $f$ $(= 2\Omega\sin\phi)$ is the Coriolis parameter. $\boldsymbol{U}$ $(= (U, V)^T$; unit: $m\ s^{-1})$ is the basic flow.



In addition to methods mentioned above, composite analysis is also used to explore the atmospheric circulation patterns related to the NAAA that cause NDJ air pollution in the NCP. The zonal Fourier harmonic analysis of atmospheric circulation is also undertaken to obtain the parameters of the atmospheric waves (van Loon et al., 1973).

## 3 Results

### 3.1 Spatial and temporal characteristics of the northeast Asian anomalous anticyclone

Figure 3 presents composite spatial distribution of atmospheric circulation for 94 NAAA events in NDJ period 1979−2021. The results show that there is a remarkably positive geopotential height anomaly at 500 hPa over Northeast Asia with a strong center, i.e., about 40ºN, 135ºE (Fig. 3a). The NCP is located in the southwest of the NAAA, which is controlled by anomalous southeasterly wind related to the NAAA (Fig. 3c). This means that the East Asian winter monsoon in the NCP, which is conduced to the accumulation of pollutants in the NCP (An et al., 2020), is weaker than normal (Wang et al., 2009). Additionally, the warm and moisture flow from the west Pacific is advected by anomalous southeasterly wind into the NCP, favoring the hygroscopic growth of pollution (Ma et al., 2014). As a result, the NCP might experience heavy air pollution weather. Significantly, the maximum of the NAAA locates about 300 hPa with a vertical structure of westward-tilt from 1000 hPa to 850 hPa (Fig. 3b). The corresponding temperature anomaly is a dipole pattern at the lower (1000 hPa to 300 hPa) and high (300 hPa to 10 hPa) level. That is to say that the lower is positive and the higher is negative temperature anomaly (Fig. 3b), which might lead to a westward-tilt structure of the NAAA via thermal wind and transient eddy feedback (Song et al., 2016).

To understand the life span of the NAAA, we show the temporal evolution of standardized daily PC time series of 94 NAAA events (Fig. 4). The PC values become positive from day −4, meaning that the NAAA starts to emerge. Note that the PC index reaches its maximum on day 0. And the PC index is almost zero or even negative from day 4, which implies the extinction of the NAAA with a life span of eight days. Moreover, the 8-day life cycle of the NAAA suggests that it is enough to investigate the intraseasonal evolution and dynamics of the NAAA in the 21-day period described in section of method. The question right now is where does the NAAA start?

To investigate the causes and evolution mechanism of the NAAA, horizontal wave activity flux is calculated and shown in the form of arrows in Fig. 5. Distinctly, there is a positive geopotential anomaly over the Gulf Stream on day −8 and propagates eastward along the upper-tropospheric polar front jet, which serves as a waveguide (Hoskins and Ambrizzi, 1993). On day −6 and the next two days, the Rossby wave energy reaches the region of Northeast Asia, but there is no positive geopotential height anomaly there. Note that the significantly positive geopotential height anomaly appears in Northeast Asia on day −3 and −2 (Fig. 5), which is an embryo of the NAAA, namely, means a rapid buildup of the NAAA. On day 0, the NAAA reaches the peak of its life cycle and wears out almost immediately on the next day (Figs. 4 and 5). There is almost





no positive geopotential height anomaly in Northeast Asia on day 4. On the interannual timescale, the NAAA seems to always occupy the whole winter and sustain degradation effect on air quality in the NCP (Chang et al., 2016; An et al., 2020). On the synoptic scale, however, the life cycle of the NAAA is just eight days. The results further suggest the necessity of studying the impact of the NAAA on air pollution in the NCP on the synoptic scale.

For a deeper understanding generation of the NAAA with a westward-tilting structure from wave theory, zonal harmonic analysis is used in this investigation. Figure 6 compares the height–longitude cross section of zonal harmonic wave anomalies on the peak day of the NAAA, overlapped with raw geopotential height anomaly. Note that the reason why other wavenumbers (i.e., wavenumbers 5−10) are not shown in Fig. 6 is that their shape are quite different from the shape of the NAAA. From Fig. 6, we find that the shape of wavenumbers 3−4 (referred as the quasi-stationary wave) is consistent with

that of the NAAA in general. The results suggest that wavenumbers 3−4 might play an important role in the generation and elimination of the NAAA (Fig. 6). The amplitudes and variances of the harmonics also support the significant roles played by the Rossby wave (Fig. 7). For instance, the amplitudes and variances of wavenumbers 3−4 are significantly greater than other wavenumbers (Fig. 7). In addition to the quasi-stationary wave characterized by wavenumbers 3−4, transient eddy feedback (2−8 days on the timescale) due to a baroclinic atmosphere also plays an important role in the development of the

NAAA (i.e., contributes to rapid buildup of the NAAA) (Song et al., 2016).

### 3.2 The northeast Asian anomalous anticyclone in relation to variation of air pollution in the NCP

Sections 3.1 investigates the spatiotemporal characteristics and evolution mechanism of the NAAA on the intraseasonal timescale, how it relates to air quality in the NCP, and what potential conditions give rise to this regime. Figure 8 presents composite $PM_{2.5}$ concentration anomaly from day −4 to 4 of 51 NAAA events in NDJ period 2000 to 2021. $PM_{2.5}$

concentration tends to increase on day −3 and then increase rapidly since day −2. By 1 day before the peak of the NAAA, $PM_{2.5}$ concentration anomaly reaches a maximum and maintains on the next day. On day 2 after the peak of the NAAA, positive $PM_{2.5}$ concentration anomaly tends to dissipate. Generally, under the background of the NAAA, the NCP experiences heavy air pollution for four days. Similarly, we investigate evolution of air pollution in the NCP based on the TAP $PM_{2.5}$ concentration and observed $PM_{2.5}$ concentration data since 2013, respectively (not shown). And results are in line

with the above conclusions drawn using TAP $PM_{2.5}$ concentration data since 2000, suggesting our finds are reliable despite $PM_{2.5}$ concentration data from machine learning by Geng et al. (2021).

Figure 9 shows daily $PM_{2.5}$ concentration anomaly averaged in the NCP for eight days before and after peak day of the NAAA. The results indicate a distinct evolution of air pollution compared with that of the NAAA. Clearly, $PM_{2.5}$ concentration begins to increase after day −4 with a peak on day −1 and then decreases gradually until zero on day 2. The

NCP has gone through significant air pollution for day −2 to 1 of the peak day of the NAAA (Fig. 9), which is consistent with the conclusions from Fig. 8. Significantly, the interquartile range (specially interdecile range) of area-averaged $PM_{2.5}$





concentration anomaly during −2 to 1 has parts less than 0, meaning that not all of the NAAA events can cause air pollution for at least three days (i.e., day −2 to 0) in the NCP. This makes us aware that the probability of air pollution events in the NCP related to the NAAA should be further examined.

To answer the question what is the probability of air pollution in the NCP caused by the NAAA in NDJ period of 2000−2021 on the intraseasonal timescale. The probability of air pollution under the background of the NAAA is presented in Table 1, and the event of air pollution is defined here as exceeding 0 for at least three days (i.e., day −2 to 0) for region-averaged $PM_{2.5}$ concentration anomaly in the NCP (Table 1). The probability of the NAAA in relation to air pollution for at least three days in the NCP is 69% if we start counting from 2000. This percent is 64% when we start counting from 2014. Additionally,

the probability of the NAAA in relation to air pollution for at least two days in the NCP is higher (i.e., 80% and 72%) than at least three days. These results further illustrate meteorological factors, especially the NAAA, play a crucial role in NDJ $PM_{2.5}$ concentration in the NCP in spite of a decline of 42% of the annual mean $PM_{2.5}$ concentrations between 2013 and 2018 in China (Cleaner air for China, 2019), which is in line with results by Dang and Liao (2019). From what is mentioned above, we come to the robust conclusion that 69% of the NAAA might cause NDJ air pollution for at least three days in the NCP

during the period of 2000−2021.

### 3.3 Why does air pollution occur in the NCP before the peak day of the NAAA

From the previous section, we see that air pollution in the NCP begins to deteriorate significantly from day −2 of the peak day of the NAAA. What sort of meteorological conditions causes this observed fact of air pollution. The NAAA is usually accompanied by southerly wind anomalies on its western flank, corresponding to lower ABLH and weaker surface winds

(Yin et al., 2017). We therefore explore the possible meteorological conditions favouring air pollution in terms of dynamics (i.e., diffusion condition) and thermodynamics (i.e., stability). In Fig. 10, the evolution of the ABLH anomaly four days before and after the peak day of the NAAA is shown. The results show that there is remarkably negative ABLH anomaly on day −3, which means a shallow atmospheric boundary layer, favourable to accumulation of pollutants. It should be noted that the ABLH reduction one day prior to the appearance of air pollution, which provides sufficient time for the accumulation of

pollutants so that the occurrence of air pollution on the following day (Figs. 8 and 11). On day 1, the negative ABLH anomaly decreases abruptly, corresponding air pollution is also lightly weakened (Figs. 8 and 11). While on the next day (i.e., day 2), there is no significantly negative ABLH anomaly, corresponding air pollution also almost disappears in the NCP (Figs. 8 and 11).

Similar conclusions can be drawn from the wind field for four days before and after the peak day of the NAAA, which

represents a diffusion condition for air pollution (e.g., Yang et al., 2016, Liu et al., 2017). As shown in Fig. 11, the NCP is mainly controlled by anomalous southerly wind with a negative divergence anomaly (not shown), which also appears one day (i.e., day −3) earlier than heavy air pollution in the NCP. The intensify and range of southerly wind increase significantly





on the following two days (i.e., day −1 and day 0). The intensify and range of southerly wind, however, shrink rapidly on day 1 and almost disappear on day 2. This process is consistent with air pollution in the NCP except that the establishment of
favourable wind field is earlier (for one day) than the occurrence of air pollution. The earlier emergence of southerly wind anomaly with a negative divergence anomaly and shallow atmospheric boundary layer together facilitate the accumulation of pollutants, leading to the happening of air pollution in the following day. And the maintenance of these two parameters leads to air pollution last for the next two days. While the later weakening and even disappearance of southerly wind anomaly and shallow atmospheric boundary layer improve air quality in the NCP after the day 1 of peak day of the NAAA.

Overall, both dynamic and thermodynamic conditions associated with the NAAA result in heavy air pollution in the NCP. Most importantly, air pollution in the NCP happens 1 day earlier than the peak of the NAAA, which provides a reference for prediction of air pollution in the NCP on the synoptic scale. In addition, if we take the positive geopotential height anomaly over domain 45ºN−60ºN, 80ºE−100ºE as a predictor, the potential prediction of air pollution in the NCP might be extended to five days (Fig. 5). Now what we wonder is why the favourable meteorological conditions related to the NAAA appear
before the peak day of the NAAA.

To further understand physical mechanisms of the NAAA favoring occurrence of air pollution in the NCP, the vertical structure of temperature and geopotential anomaly and their evolution in position of 37ºN, 115ºE are shown in this section. As shown in Fig. 12, temperature anomaly features a backward tilt with height below 700 hPa, meaning that the higher temperature anomaly moves from higher to lower level from day −3 to −1, which is easier to cause a potential thermal
inversion. In addition, the ABLH anomaly is significantly negative in this period (Fig. 10 and 12), corresponding negative $PM_{2.5}$ concentration anomaly in the NCP (Fig. 8). While above 700 hPa, it features a forward tilt with height, implying that the positive temperature anomaly moves form lower to higher level from day −1 to 2, which is unfavourable for the formation of a potential thermal inversion. Besides, there is significantly anomalous ascending motion in the troposphere on day −3 to −1 (not shown), which might suppress intrusions of clean air from upper levels (i.e., above 300 hPa) to the lower
levels, resulting in a shallower atmospheric boundary layer (Zhong et al., 2019). However, the negative ABLH anomaly decreases rapidly from day −1 to 2, and corresponding air quality in the NCP is gradually improved (Fig. 10 and 12). The characteristics based on area-averaged temperature and ABLH anomaly in the NCP are similar to the results based on single position (not shown).

Similarly, we check the evolution of geopotential height anomaly and meridional wind anomaly with time and height in
point (37ºN, 115ºE). Unsurprisingly, positive geopotential height anomaly shows a sudden enhancement in the whole troposphere from day −4 to 3 (Fig. 13), which is in line with result in Fig. 4b. This means a rapid buildup of the NAAA with a sudden enhancement of anomalous southerly wind. As we all know, air pollution is closely tied to lower wind field, especially surface wind field (e.g., Yang et al., 2016; Liu et al., 2017; Yin et al., 2017). Therefore, on day −3, the NAAA rapidly builds with the sudden increase of anomalous geopotential height anomaly and southerly wind anomaly (Figs. 7 and



13), resulting in air pollution prior to the peak day of the NAAA. We can draw the same conclusion compared with an area-averaged geopotential height anomaly and 1000 hPa meridional wind anomaly in the NCP.

## 4 Conclusions and discussion

In this study, we investigate the characteristics and evolution mechanisms of the NAAA on the intraseasonal timescale and the associated air pollution in the NCP in NDJ. In particular, the intraseasonal NAAA has a life span of about eight days with
a structure of westward tilt with height, and its evolution is closely tied to the Rossby wave from upstream (i.e., the North Atlantic). On day −8, there is significant circulation anomaly over the Gulf Stream and downstream propagation in the form of the Rossby wave. The NAAA reaches its peak on day 0 and decreases rapidly the next day. According to harmonic analysis, the NAAA with a westward tilt may be related to the wavenumbers 3−4. Additionally, the NAAA is also enhanced by the transient eddy, which can be induced by weak baroclinic atmosphere with the characteristic of vertical dipole pattern
of temperature. For instance, Song et al. (2016) found that the transient eddy feedback leads to 30% of the NAAA amplification using geopotential height tendency equation.

Further results show that 69% of the NAAA in NDJ period of 2000−2021 causes regional air pollution for at least three days (80% for at least two days) in the NCP and its peak day lags occurrence of air pollution for two days. The composite analysis reveals that the shallower atmospheric boundary layer and stronger surface southerly wind anomaly (weaker northerly wind)
associated with the NAAA in the NCP appear one day prior to air pollution, which provides dynamic and thermal conditions for the accumulation of pollutants and finally occurrence of air pollution on the following day. We also find that the stagnant air and weak ventilation conditions are determined by a special vertical distribution of temperature anomaly and a rapid buildup of the NAAA.

In addition, it is well known that the wet deposition through scavenging by rainfall is an effective way to remove
atmospheric aerosols and soluble gases (e.g., Atlas and Giam, 1988). When the NAAA appears, southern China tends to experience heavy rainfall and vice versa in the NCP (not shown) (e.g., Ma et al., 2018; An et al., 2020, 2021), which is only not conducive to the wet removal of aerosol in the NCP, but usually deteriorates air quality in the NCP via a local north−south circulation (not shown) (An et al., 2020, 2021; Mei et al., 2021).

In brief, the NAAA and associated meteorological parameters play a crucial role in formation of NDJ air pollution in the
NCP on the intraseasonal timescale, which is slightly different from its role on wintertime air pollution in this region on the interannual timescale. For example, we can't draw a conclusion that the peak time of the NAAA lags air pollution in the NCP in NDJ on the interannual timescale. In addition, there is usually an anomalous descending motion in the NCP in NDJ on the interannual timescale (An et al., 2020), while on the intraseasonal timescale is an anomalous ascending motion in this region (not shown). The shortcoming of this study only investigates the influence of the NAAA on air pollution in the NCP in NDJ





on the intraseasonal timescale. It should be noticed that cyclone anomaly in Northeast Asia, as a pattern of out-of-phase of
the NAAA (Wang et al., 2009; Song et al., 2016), might be a favorable atmospheric circulation to improve air quality, which
should also be studied in future.

**Code availability**

Codes used in this paper are available upon request to the corresponding author.

**Data availability**

NCEP reanalysis 2 data provided by the NOAA/OAR/ESRL PSL, Boulder, Colorado, USA, from their Web site at
https://psl.noaa.gov/data/gridded/data.ncep.reanalysis2.html.
Daily boundary layer height data is available from https://cds.climate.copernicus.eu/cdsapp#!/home.
The hourly $PM_{2.5}$ concentration data can be downloaded from https://quotsoft.net/air/ and http://tapdata.org.cn/ (TAP).

**Author contributions:** XA, LS and WC designed the experiments and carried them out. XA downloaded and analysed the
reanalysis data and prepared all the figures. XD prepared the manuscript with contributions from all co-authors. LS, WC, PH,
and SC revised the manuscript.

**Competing interests:** The authors declare that they have no conflict of interest.

**Acknowledgements**

This research was supported by the National Natural Science Foundation of China (grant no. 41975008).

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





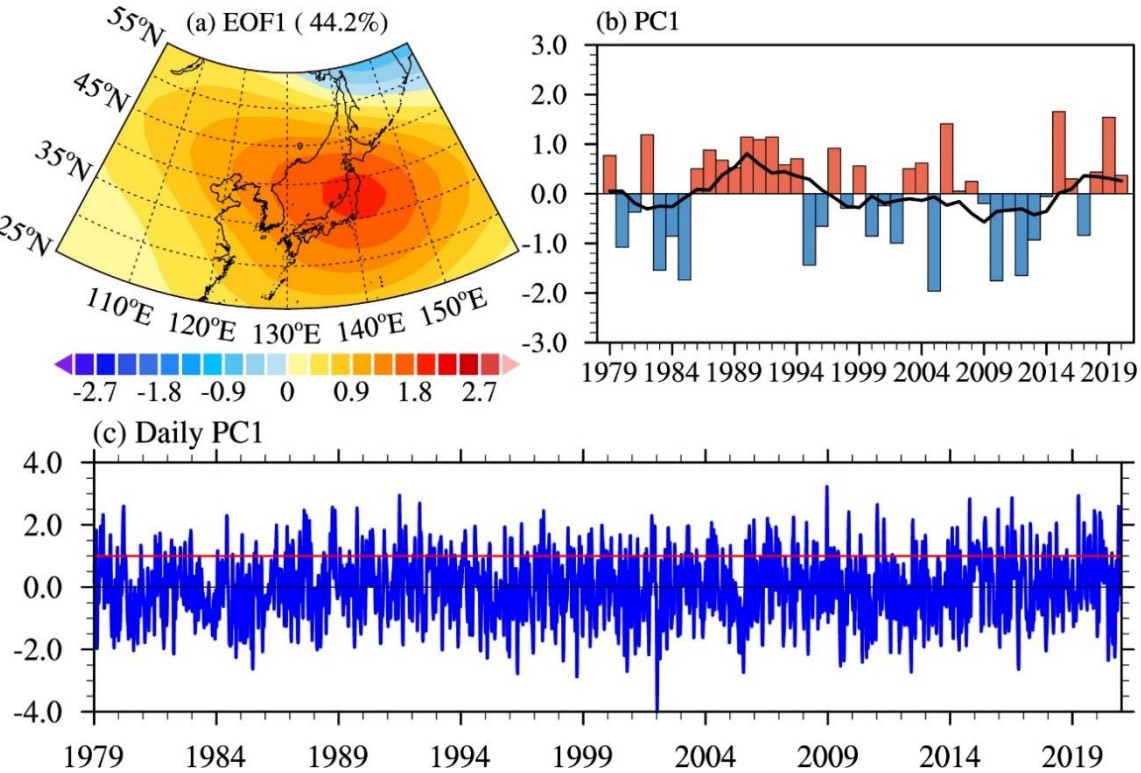

**Figure 1: (a) The spatial pattern of the first EOF and (b) the corresponding standardized PC series over the domain 25ºN–55ºN, 100ºE–160ºE in NDJ period 1979–2020. (c) The standardized daily PC series in NDJ period 1979–2020, red line is one standard deviation.**





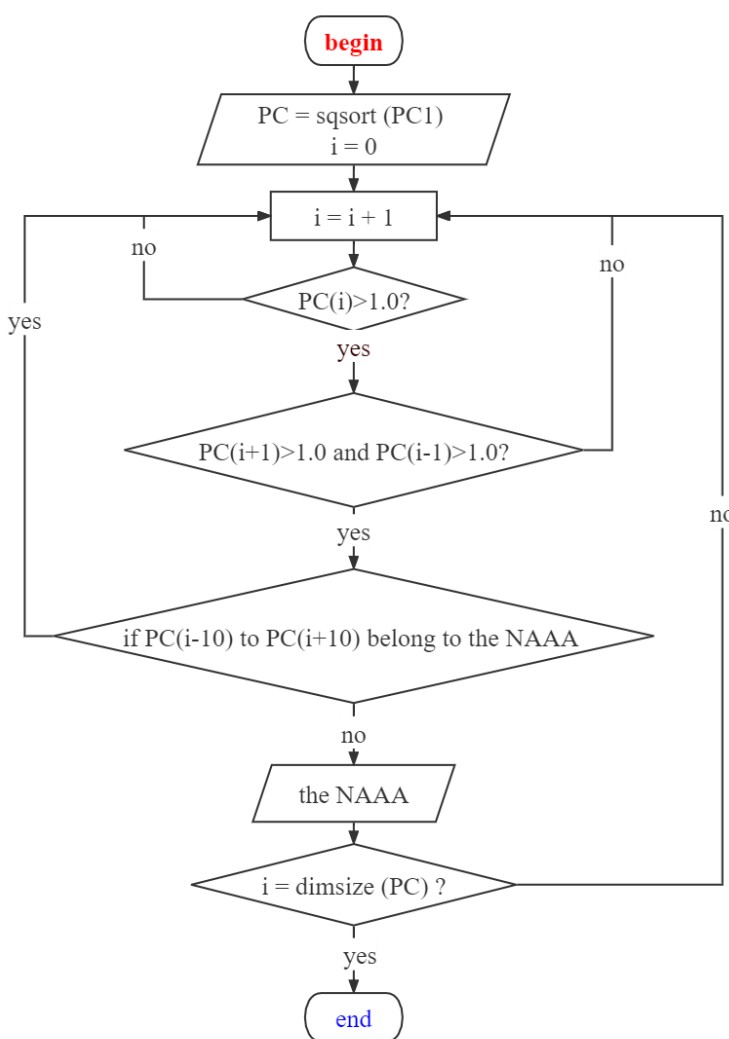

**Figure 2: Process for selecting the NAAA events. Here, PC is daily index of the NAAA, sqsort (PC) represents PC is sorted in descending order, dimsize (PC) is the size of one-dimensional array PC.**





**Figure 3: (a) Composite geopotential height anomaly (shaded and contours; unit: m) at 500 hPa on the peak day of 94 NAAA events in NDJ period 1979 to 2021. (b) Composite longitude-height cross section of geopotential height anomaly (shaded; unit: m) and temperature anomaly (contours; unit: ºC) at 37ºN. (c) Composite wind vector (arrows; unit: m s⁻¹) and wind speed (shaded; unit: m s⁻¹) at 850 hPa. The white dots denote the 99% confidence level according to the Student's *t* test.**





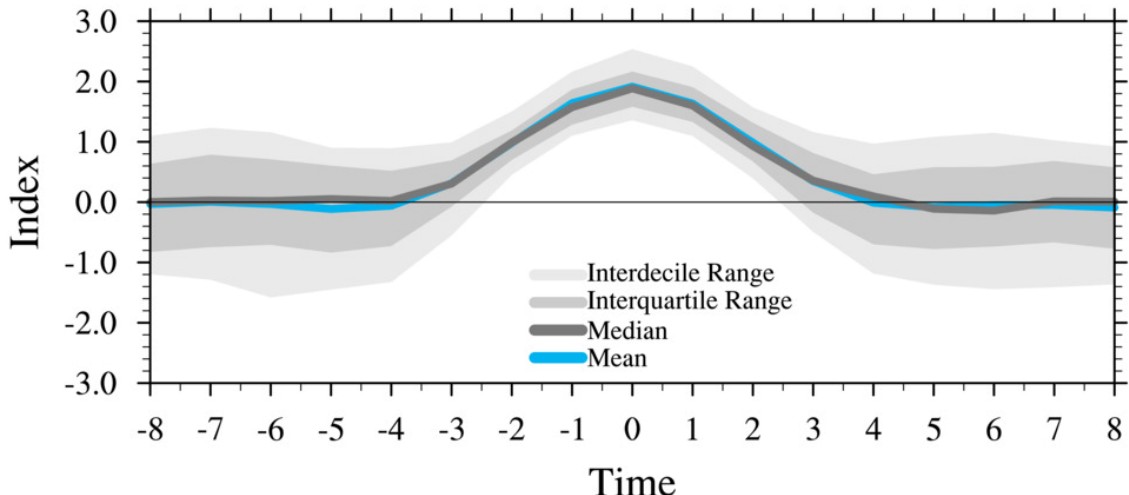

**Figure 4: The temporal evolution of the standard PC time series for 94 NAAA events. In particular, black and blue curves, light**

**and deep gray filling represent mean and median value, interdecile range and interquartile range, respectively.**





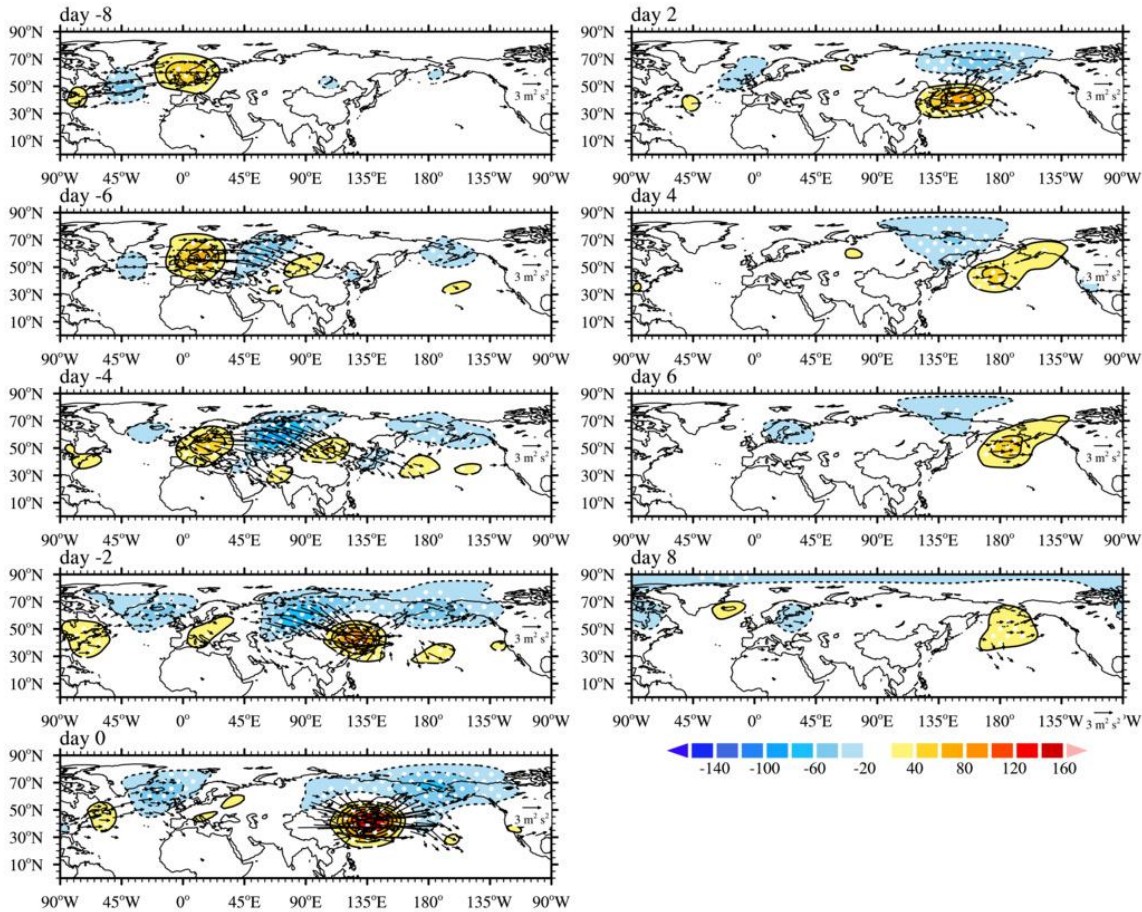

**Figure 5: Composite evolution of geopotential height anomaly (shaded; unit: m) and the WAF (arrows; unit: m² s²) at 500 hPa on days −8, −6, −4, −2, 0, +2, +4, +6 and +8 in 94 NAAA events since 1979. The white dots denote the 99% confidence level according to the Student's *t* test.**





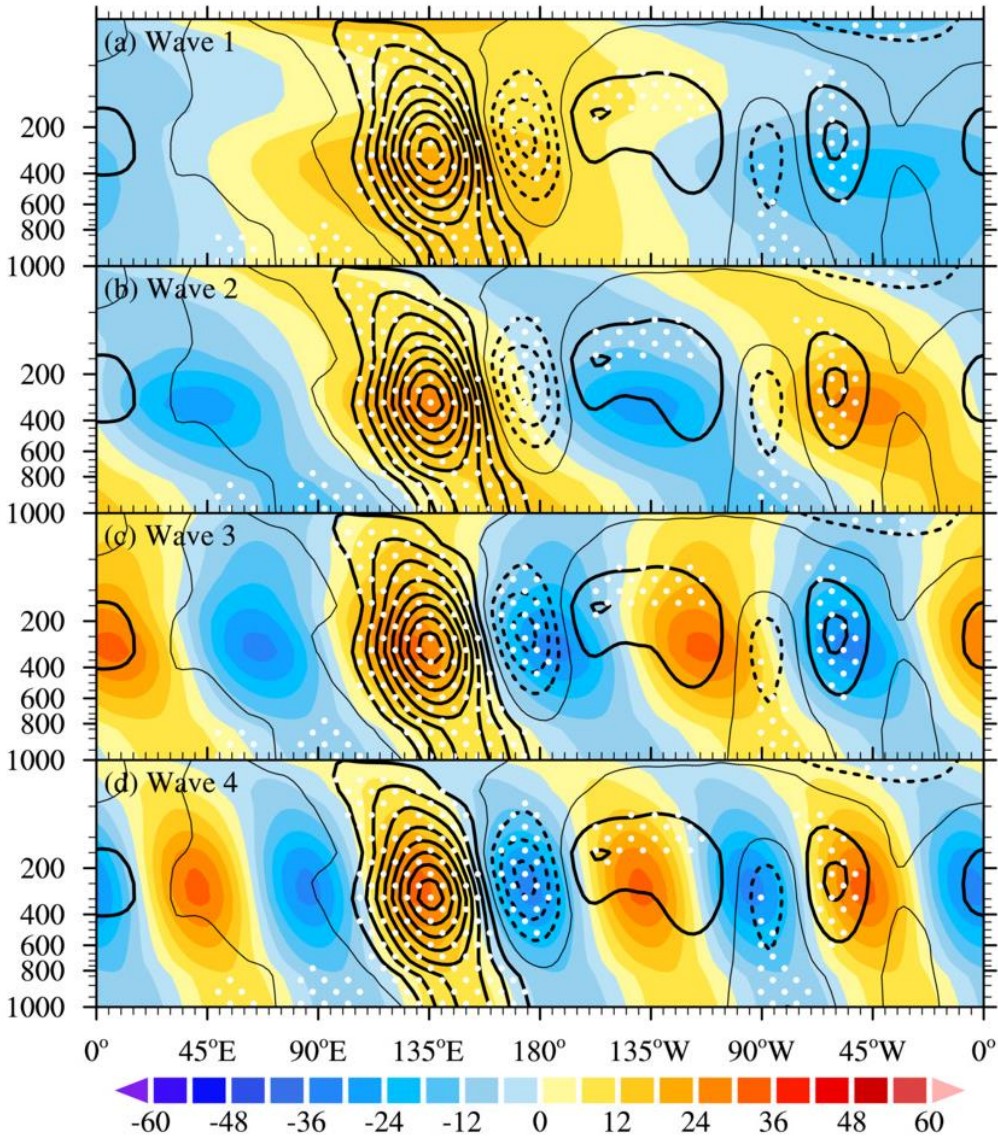

**Figure 6: Cross section of zonal harmonic analysis of geopotential height anomaly. (a) wavenumber 1, (b) wavenumber 2, (c) wavenumber 3 and (d) wavenumber 4 (shaded; unit: m) and geopotential height anomaly (contours; unit: m) from 1000 hPa to 10 hPa on peak day of 94 NAAA events. The white dots denote the 99% confidence level according to the Student's *t* test.**





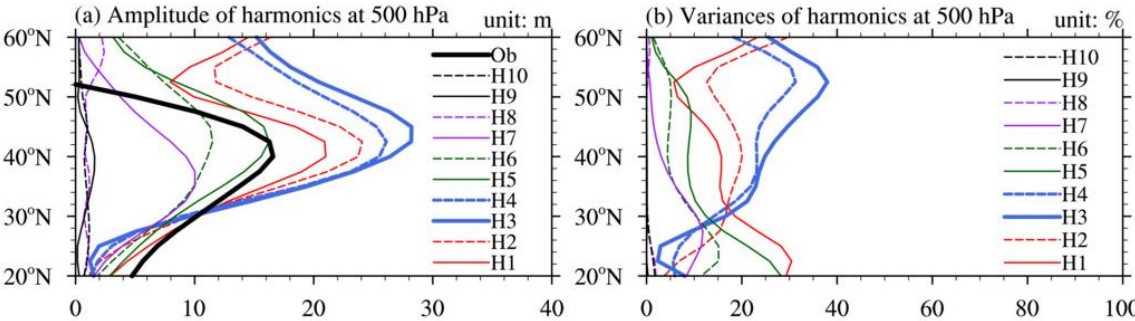

**Figure 7: Zonal harmonic analysis of geopotential height anomaly at 500 hPa on the peak day of 94 NAAA events since 1979. (a) Amplitude harmonics of wave 1 to wave 10, (b) variances harmonics of wavenumber 1 to 10. In particular, the thick black curve (Ob) represents zonal mean geopotential height anomaly in the peak day of 94 NAAA events.**







**Figure 8: Composite evolution of PM$_{2.5}$ concentration anomaly (shaded; unit: ug m$^{-3}$) from day -8 to day 8 in 51 NAAA events in NDJ period 2000−2021. The white dots denote the 99% confidence level according to the Student's *t* test.**




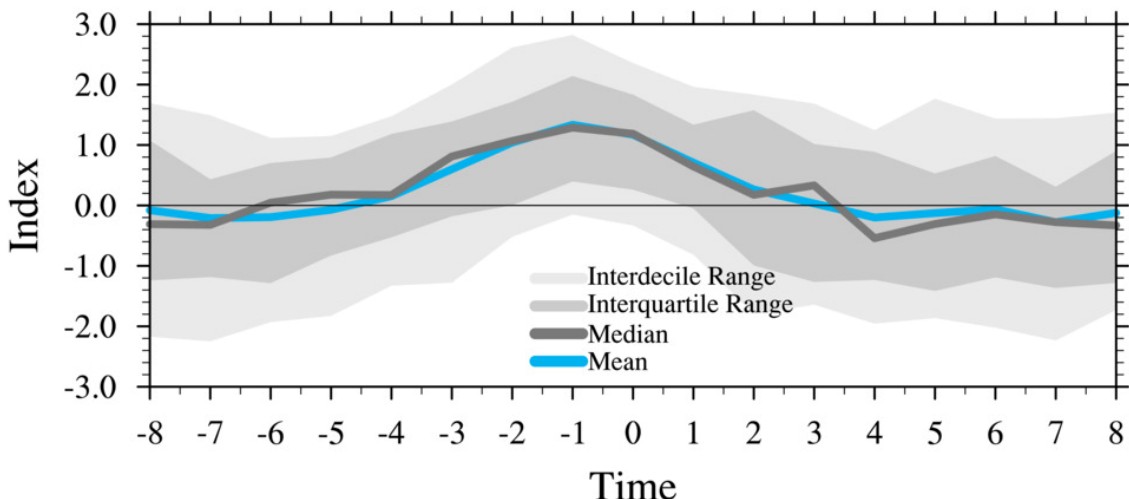

**Figure 9: The same as Fig. 3 except for PM₂.₅ concentration anomaly in 51 NAAA events in NDJ period 2000−2021.**





**Figure 10: The same as Fig. 5 except for the ABLH anomaly (shaded; unit: m).**







**Figure 11: The same as Fig. 5 except for anomalous wind vector (arrows; unit: m s⁻¹) and anomalous wind speed (shaded; unit: m s⁻¹) at 1000 hPa.**






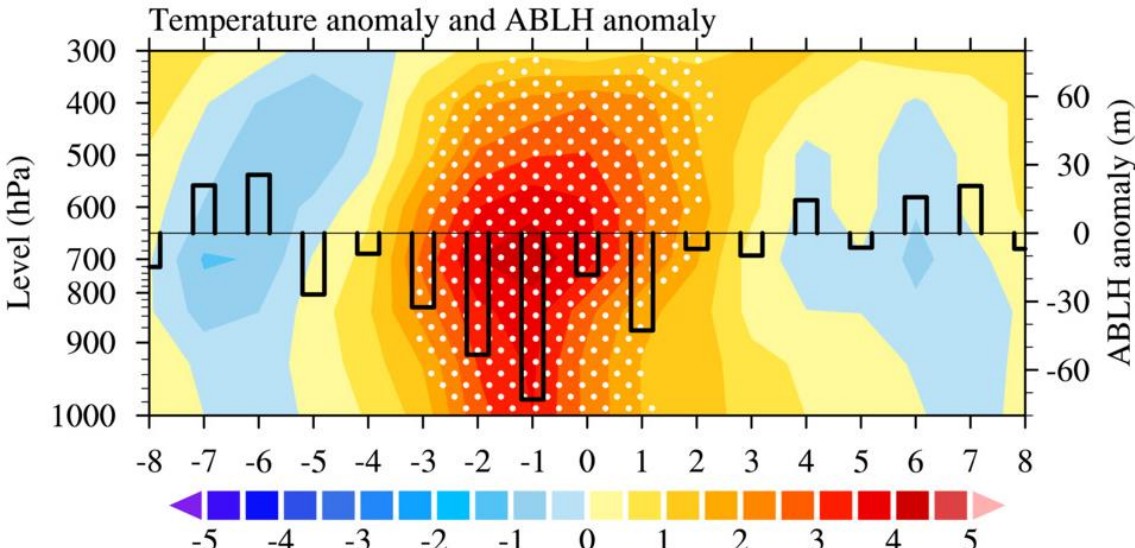

Figure 12: Composite time-height cross section of temperature anomaly (shaded; unit: m) at 37ºN, 115ºE from the developing stage (day –8 to day 0) to the decaying stage (day 0 to day +8) for 51 NAAA events in NDJ period 2000−2021. The white dots denote the 99% confidence level according to the Student's *t* test. Black bar charts represent the variation of the ABLH anomaly

(bars; unit: m) at 37ºN, 115ºE from the developing stage (day –8 to day 0) to the decaying stage (day 0 to day +8) of 51 NAAA events in NDJ period 2000−2021. Especially, the coordinates on the right represent the value of the ABLH anomaly. The horizontal line denotes zero of the ABLH anomaly.


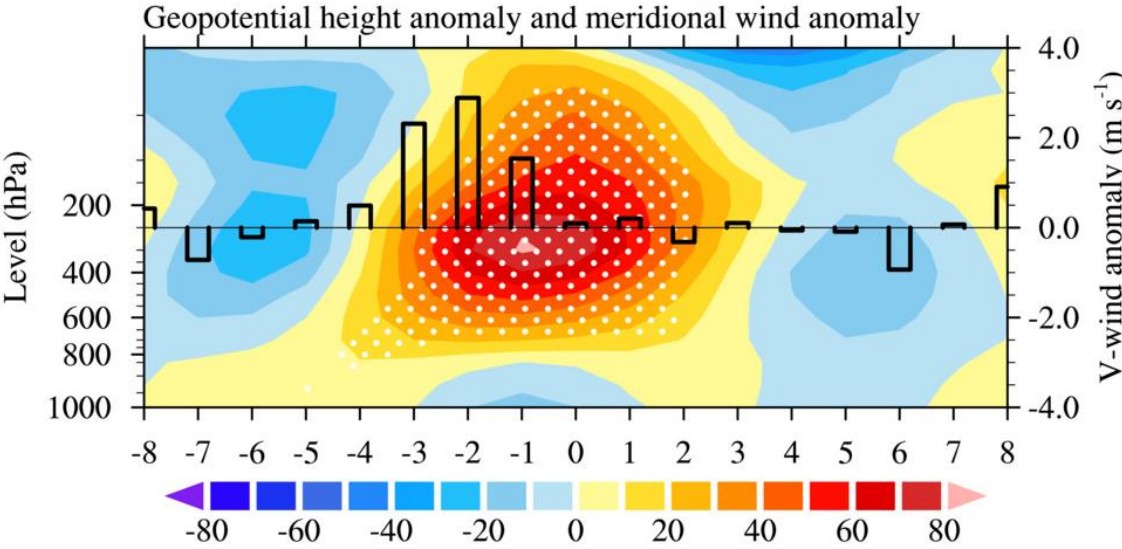

Figure 13: The same as Fig. 9, but for geopotential height anomaly (shaded; unit: m) and 1000-hPa meridional wind anomaly (bars; unit: m s⁻¹). Positive values of meridional wind anomaly represent southerly wind anomaly.





**Table 1: The probability of the NAAA in relation to air pollution in the NCP.**

|  | The NAAA | The number of air pollution lasted two days (day −1 and day 0) | | The number of air pollution lasted three days (day −2 and day 0) | |
| --- | --- | --- | --- | --- | --- |
|  | number | number | percent | number | percent |
| 2000.11–2021.01 | 51 | 41 | 80% | 35 | 69% |
| 2014.11–2021.01 | 25 | 18 | 72% | 16 | 64% |