# Peer review of "Intraseasonal variation of the northeast Asian anomalous anticyclone and its impacts on $PM_{2.5}$ pollution in the North China Plain in early winter"

_Atmospheric Chemistry and Physics, 2021_

## Referee Comment (RC1)

This study investigated the important impacts of the northeast Asian anomalous anticyclone (NAAA) on the intraseasonal variations of PM2.5 pollution in the North China Plain (NCP). The paper presents novel concepts, ideas and tools. The scientific methods and assumptions are valid and clearly outlined so that substantial conclusions are reached. The description of dataset and calculations are sufficiently complete and precise. Hence, the manuscript is recommended for publication after minor revision. Specific comments can be found as follows.

General comments:

In this study, the authors reported the air quality deterioration two day prior to the peak day of NAAA and suggested that the geopotential height anomaly and meridional wind anomaly were the main causes. However, the cause and effect between the anomalies and PM2.5 should be fully discussed. Because, increases in aerosols, especially absorbing aerosols, can also heat the air, leading to the atmospheric stagnation (e.g., Ding et al., 2016), and decrease the winds over the NCP (e.g., Lou et al., 2019). Therefore, the aerosol accumulation may also cause or at least intensify the dynamical or thermodynamical anomalies.

Specific comments

Title: The term "air pollution" was used throughout the text including the title. But this study only focused on $PM_{2.5}$ pollution. It should be revised for the whole text.

Lines 16 and 21: "day -3" and day "day -1" should be made clearer.

Line 39: All sea surface temperature studies listed here are belong to ENSO impacts. It could be more accurately presented. Also, a recent study also reveal impacts of different duration of El Nino on PM2.5 over China (Zeng et al., 2021).

Line 43: Studies also revealed that the aerosol pollution over NCP during COVID-19 was related to NAAA, which could be included here (Ren et al., 2021).

Line 77: The $PM_{2.5}$ data used in this study were from TAP generated using a machine learning approach and covers the period from 2000 to present. But the reanalysis data are from 1979 to present. The authors is suggested to use the long-term aerosol data for the same period as meteorological parameters in future studies (e.g., Li et al., 2021).

Line 100: What do the 'e' and 'T' mean?

Line 125: "which is conduced to the accumulation of pollutants in the NCP" should be given after "is weaker than normal".

Line 154: Please give a short description about the wavenumbers 1-10.

References:

Ding, A. J., Huang, X., Nie, W., Sun, J. N., Kerminen, V. M., Petäjä, T., et al. (2016). Enhanced haze pollution by black carbon in megacities in China. Geophysical Research Letters, 43, 2873–2879. https://doi.org/10.1002/2016GL067745.

Lou, S., Yang, Y., Wang, H., Smith, S. J., Qian, Y., & Rasch, P. J. (2019). Black carbon amplifies haze over the North China Plain by weakening the East Asian winter monsoon. Geophysical Research Letters, 46, 452–460. https://doi.org/10.1029/2018GL080941.

Zeng, L., Yang, Y., Wang, H., Wang, J., Li, J., Ren, L., Li, H., Zhou, Y., Wang, P., and Liao, H.: Intensified modulation of winter aerosol pollution in China by El Niño with short duration, Atmos. Chem. Phys., 21, 10745–10761, https://doi.org/10.5194/acp-21-10745-2021, 2021.

Ren, L., Yang, Y., Wang, H., Wang, P., Chen, L., Zhu, J., and Liao, H.: Aerosol transport pathways and source attribution in China during the COVID-19 outbreak, Atmos. Chem. Phys., 21, 15431–15445, https://doi.org/10.5194/acp-21-15431-2021, 2021.

Li, H., Yang, Y., Wang, H., Li, B., Wang, P., Li, J., and Liao, H., Constructing a spatiotemporally coherent long-term PM2.5 concentration dataset over China during 1980–2019 using a machine learning approach, Sci. Total Environ., 765, 144263, https://doi.org/10.1016/j.scitotenv.2020.144263, 2021.

---

## Author Response (AR1)

Dear editor,

We appreciate you and two reviewers for carefully reviewing our manuscript and providing the valuable suggestions to improve our paper. We have carefully read all comments and revised the manuscript as suggested. The following are our responses to all comments point by point. The italicized sentences are all comments, and the other sentences are the author's responses. The green sentences and words are the specific revisions for anonymous reviewer #1. The blue sentences and words are the specific revisions for anonymous reviewer #2. We also marked all relevant changes in the manuscript in the same way.

To reviewer #1,

Dear Reviewer,

Thank you very much for your comments on this paper. We have carefully read all comments and revised the manuscript as suggested. The following are our responses to all comments point by point. The italicized sentences are all comments, and the other sentences are the author's responses. The green sentences and words are the specific revisions. We also marked all relevant changes in the manuscript in the same way.

*This study investigated the important impacts of the northeast Asian anomalous anticyclone (NAAA) on the intraseasonal variations of PM2.5 pollution in the North China Plain (NCP). The paper presents novel concepts, ideas and tools. The scientific methods and assumptions are valid and clearly outlined so that substantial conclusions are reached. The description of dataset and calculations are sufficiently complete and precise. Hence, the manuscript is recommended for publication after minor revision. Specific comments can be found as follows.*

***General comments:***

*(1) In this study, the authors reported the air quality deterioration two day prior to the peak day of NAAA and suggested that the geopotential height anomaly and meridional wind anomaly were the main causes. However, the cause and effect between the anomalies and PM2.5 should be fully discussed. Because, increases in aerosols, especially absorbing aerosols, can also heat the air, leading to the atmospheric stagnation (e.g., Ding et al., 2016), and decrease the winds over the NCP (e.g., Lou et al., 2019). Therefore, the aerosol accumulation may also cause or at least intensify the dynamical or thermodynamical anomalies.*

**Response:** Thank you so much for your suggestion.

Following your advice, we have added a discussion about the effect of $PM_{2.5}$ on the circulation anomalies in Section 4 of the revised manuscript.

**Lines 266−272:** "In addition, as shown in Figs. 8−9, PM2.5 pollution occurred over the NCP one day before the peak

day of the NAAA, which implies the former might exert an impact on its formation and thus form a positive feedback of PM2.5 pollution−atmospheric circulation. The increases in aerosols, especially absorbing aerosols, have been reported to heat the air, and therefore lead to the atmospheric stagnation (e.g., Ding et al., 2016) and the weaker ventilation over the NCP (e.g., Lou et al., 2019). Therefore, the PM2.5 accumulation before the peak day of the NAAA may also cause or at least intensify the dynamical or thermodynamical anomalies, which in turn might support the formation of PM2.5 pollution over the NCP. This is a potential interesting topic that deserves further investigation in the future."

*Specific comments:*

*1. Title: The term "air pollution" was used throughout the text including the title. But this study only focused on $PM_{2.5}$ pollution. It should be revised for the whole text.*

Response: Thank you for your valuable idea for this paper. We have revised this description in the whole text. So please check the revised manuscript.

*2. Lines 16 and 21: "day -3" and day "day -1" should be made clearer.*

Response: Thank you for your good suggestion. We have revised these descriptions in the manuscript.

**Line 17:** "… day −3 …" −> "… day 3 prior to its peak day …"

**Line 22:** "… day −1 …" −> "… the next day …"

*3. Line 39: All sea surface temperature studies listed here are belong to ENSO impacts. It could be more accurately presented. Also, a recent study also reveals impacts of different duration of El Nino on $PM_{2.5}$ over China (Zeng et al., 2021).*

Response: Thank you for your carefully check and help. We have revised this sentence as suggested, so please check the revised manuscript.

**Lines 40−41:** "… including El Niño (Chang et al., 2016; Jeong et al., 2018; Yu et al., 2020; Zeng et al., 2021), …"

*4. Line 43: Studies also revealed that the aerosol pollution over NCP during COVID-19 was related to NAAA, which could be included here (Ren et al., 2021).*

Response: Thank you for your carefully check and help. We have revised this sentence as suggested, so please check the revised manuscript.

**Lines 43−45:** "Studies also revealed that the aerosol pollution over the NCP during COVID-19 was related to the northeast Asian anomalous anticyclone (NAAA) (Ren et al., 2021)."

*5. Line 77: The $PM_{2.5}$ data used in this study were from TAP generated using a machine learning approach and covers*

*the period from 2000 to present. But the reanalysis data are from 1979 to present. The authors are suggested to use the long-term aerosol data for the same period as meteorological parameters in future studies (e.g., Li et al., 2021).*

**Response:** Thank you for your suggestion. We will use this data for future studies as much as possible. In fact, it has been used to examine air pollution in our other papers. $PM_{2.5}$ data provided by Yang (2020) is in excellent agreement with ground measurements, and can capture a trend of continuous increase in the mean $PM_{2.5}$ concentrations from 1985 to 2014 in China (Li et al., 2021), even daily evolution characteristics of $PM_{2.5}$ over the North China Plain (An et al., 2022).

*6. Line 100: What do the 'e' and 'T' mean?*

**Response:** We are sorry for this confusion. "*e*" represents the spatial pattern of EOF1, which is mentioned in Line 100 (i.e., the EOF1 spatial pattern (e)). We're sorry we didn't use italics for e. $e^T$ is called the transpose of *e*. We have added these descriptions into the manuscript.

**Line 99:** "… the EOF1 spatial pattern (*e*) …"

**Line 102:** "Here, $e^T$ is called the transpose of *e*."

*7. Line 125: "which is conduced to the accumulation of pollutants in the NCP" should be given after "is weaker than normal".*

**Response:** Thank you for your carefully check and suggestion. We have revised this sentence.

**Lines 127−128:** "… is weaker than normal (Wang et al., 2009), which is conduced to the accumulation of pollutants in the NCP (An et al., 2020)."

*8. Line 154: Please give a short description about the wavenumbers 1-10.*

**Response:** Thank you for your valuable advice. We have added the description of the wavenumbers 1-10 in Section 3.1 of the revised manuscript.

**Lines 153−155:** "Wavenumbers 1−10 are the spectral components on wave-number domain produced by Fourier transform in the spatial domain. Among them, wavenumbers 1−2 represent ultra-long wave, wavenumbers 3−5 denote long wave, and wavenumbers 6−10 are synoptic waves."

**References**

An, X. D., Sheng, L. F., Li, C., Chen, W., Tang, Y. L., and Huangfu, J. L.: Effect of rainfall-induced diabatic heating over southern China on the formation of wintertime haze on the North China Plain, Atmos. Chem. Phys., 22, 725–738, https://doi.org/10.5194/acp-22-725-2022, 2022.

Ding, A. J., Huang, X., Nie, W., Sun, J. N., Kerminen, V. M., Petäjä, T., et al.: Enhanced haze pollution by black carbon in

megacities in China, Geophys. Res. Lett., 43, 2873–2879, https://doi.org/10.1002/2016GL067745, 2016.

Lou, S., Yang, Y., Wang, H., Smith, S. J., Qian, Y., and Rasch, P. J.: Black carbon amplifies haze over the North China Plain by weakening the East Asian winter monsoon, Geophys. Res. Lett., 46, 452–460, https://doi.org/10.1029/2018GL080941, 2019.

Zeng, L., Yang, Y., Wang, H., Wang, J., Li, J., Ren, L., Li, H., Zhou, Y., Wang, P., and Liao, H.: Intensified modulation of winter aerosol pollution in China by El Niño with short duration, Atmos. Chem. Phys., 21, 10745–10761, https://doi.org/10.5194/acp-21-10745-2021, 2021.

Ren, L., Yang, Y., Wang, H., Wang, P., Chen, L., Zhu, J., and Liao, H.: Aerosol transport pathways and source attribution in China during the COVID-19 outbreak, Atmos. Chem. Phys., 21, 15431–15445, https://doi.org/10.5194/acp-21-15431-2021, 2021.

Li, H., Yang, Y., Wang, H., Li, B., Wang, P., Li, J., and Liao, H.: Constructing a spatiotemporally coherent long-term PM2.5 concentration dataset over China during 1980–2019 using a machine learning approach, Sci. Total Environ., 765, 144263, https://doi.org/10.1016/j.scitotenv.2020.144263, 2021.

Dear Reviewer,

We appreciate you for carefully reviewing our manuscript and providing the valuable suggestions to improve our paper.

5 We have carefully read all comments and revised the manuscript as suggested. The following are our responses to all comments point by point. The italicized sentences are all comments, and the other sentences are the authors' responses. The blue sentences and words are the specific revisions. We also marked all relevant changes in the manuscript in the same way.

*In this study, the authors investigated the impacts of the northeast Asian anomalous anticyclone (NAAA) on the PM2.5*

10 *concentrations in the North China Plain (NCP) on the intraseasonal timescale. The authors found that, under NAAA condition, the probability of regional air pollution for at least three days in the NCP is 69% in NDJ (November to January) period 2000-2021. The manuscript is well written, and results are clearly presented. I have a few comments for the authors to consider.*

***General Comments:***

*(1) The authors mentioned in the abstract that there is considerable intraseasonal variability of NAAA in NDJ period.*

15 *One of the objectives for this study is to derive the characteristics of air pollution evolution in the NCP under the background of the NAAA in NDJ on the intraseasonal timescale. It is a little unclear to me that what the authors mean for intraseasonal variability. Is it for temporal evolution (up to 17 days) of the PC time series for 94 NAAA events and evolution of PM2.5 concentration anomaly? If yes, it seems more like synoptic/weather scale to me.*

**Response:** We are sorry for the confusion. The intraseasonal variability is for temporal evolution (up to 17 days) of 94

20 NAAA events instead of evolution of $PM_{2.5}$ concentration anomaly. In this manuscript, we have filtered out high-frequency signal (i.e., synoptic scale) using 8–90-day Butterworth bandpass-filter for the PC time-series of the NAAA (Fig. 1c), which thus reflects the intraseasonal variability of the NAAA events instead of synoptic-scale variability. Song et al. (2016) also extracted the intraseasonal variability of East Asian trough using 8–90-day Butterworth bandpass-filter and found the intraseasonal East Asian trough events have a life span of 18 days. In addition, the range of synoptic scale is usually one week

25 approximately (Fig. S1). In the current manuscript, the time-scale range of the NAAA is up to 17 days (more than 2 weeks), exceeding the synoptic scale (i.e., one week) (Figs. S1−S2). However, it is synoptic variability if it is only for the temporal evolution of the $PM_{2.5}$ concentration anomaly. In other words, we find that the intraseasonal NAAA events can cause air pollution process (3−7 days). In any case, the definition of the time scale for NAAA events will not change the results of this study. Thank you again for your advice.

[Figure]

**Figure S1: Processes that act as sources of ISI climate predictability extend over a wide range of timescales and involve interactions among the atmosphere, ocean, and land (National Research Council, 2010). For the y-axis, "A" indicates "atmosphere;" "L" indicates "land;" "I" indicates "ice;" and, "O" indicates "ocean."**

[Figure]

**Figure S2: Schematic depiction of (bottom) temporal ranges and (top) sources of predictability for weather and climate prediction (Merryfield et al., 2020).**

*(2) I wonder how many NAAA events of the selected 94 events occurred in November and January. Are these events evenly distributed in the three months? Are there any differences for the characteristics of NAAA in the three months, like time duration*

*and strength? Is the probability of regional air pollution under NAAA similar for all three months in NDJ? There are some similar composite analyses from Zhong et al. (2019) for December. I wonder if there are any notable/important differences for the impacts of NAAA on the air pollution among the three months. Or the inclusion of November and January is just for more NAAA events?*

5     **Response:** We are sorry for this confusion. The NAAA events during November to January were selected because air pollution over the NCP occurs frequently in these months (Wang et al., 2019; An et al., 2020; Yin et al., 2021). Furthermore, atmosphere circulations leading to haze events might be consistent in December and January (Yin et al., 2021). Of course, there are more NAAA events if November and January are included, compared to that by Zhong et al. (2019). Figure S3 displays the distribution of geopotential height anomalies at 500 hPa in November, December and January. The characteristics

10    of NAAA events in these three months are similar (Fig. S3). In addition, the number of NAAA events in November, December and January is 26, 31 and 39 respectively. If the NAAA are selected during 2000−2021, the number of NAAA events is 15, 16 and 20 in November, December and January, and associated probability of $PM_{2.5}$ pollution (at least two days) is 67%, 94% and 80%, respectively. From the current results, NAAA events in December seem to be more conducive to $PM_{2.5}$ pollution over the NCP. These interesting results need to be studied in the future for physical mechanisms.

[Figure]

15

**Figure S3: Composite geopotential height anomaly (shaded and contours; unit: m) at 500 hPa on the peak day of the**

**NAAA events in (a) November, (b) December, and (c) January during 1979 to 2021. The white dots denote the 99% confidence level according to the Student's *t* test.**

**Table S1: The number and probability of NAAA events in November, December and January, respectively.**

| | November | December | January | Total |
|---|---|---|---|---|
| NAAA number during 1979.11−2021.01 | 26 | 31 | 37 | 94 |
| (NAAA number of each month/total NAAA number of NDJ) | (28%) | (33%) | (39%) | |
| NAAA number during 2000.11−2021.01 | 15 | 16 | 20 | 51 |
| (NAAA number of each month/total NAAA number of NDJ) | (29%) | (31%) | (39%) | |
| Number of PM$_{2.5}$ pollution lasted two days (day –1 and day 0) during 2000.11−2021.01 | 10 | 15 | 16 | 41 |
| (pollution number of each month/NAAA number of each month) | (67%) | (94%) | (80%) | (80%) |

*(3) It is interesting that air pollution occurs in the NCO one day before the peak day of the NAAA. I agree with the other reviewer that the cause and effect between NAAA and PM2.5 pollution should be fully discussed.*

**Response:** Thank you for your valuable suggestion. We agree with you. The NAAA is important to the formation of PM$_{2.5}$ pollution over the NCP, and the impacts of PM$_{2.5}$ on the NAAA is not to be ignored. There is evidence that the changes in aerosols can cause the response of atmospheric circulation (Lou et al., 2019). For example, previous studies revealed that strong aerosol-PBL interaction during the polluted period results in a suppressed and stabilized PBL and elevated humidity, triggering a positive feedback to amplify the haze severity at the ground level (Lin et al., 2021). However, in the current manuscript, we focus on the roles of atmospheric circulation (i.e., the NAAA) on the formation of PM$_{2.5}$ pollution over the NCP. But undeniably, the effect of aerosols (i.e., PM$_{2.5}$) on atmospheric circulations is worthy of further study. Studies on the effect of PM$_{2.5}$ on the NAAA will might be carried out in our future paper. To show the importance of this issue, we have added some discussion in Section 4 of the manuscript.

**Lines 266−272:** "In addition, as shown in Figs. 8−9, PM2.5 pollution occurred over the NCP one day before the peak day of the NAAA, which implies the former might exert an impact on its formation and thus form a positive feedback of PM2.5 pollution−atmospheric circulation. The increases in aerosols, especially absorbing aerosols, have been reported to heat the air, and therefore lead to the atmospheric stagnation (e.g., Ding et al., 2016) and the weaker ventilation over the NCP (e.g., Lou et al., 2019). Therefore, the PM2.5 accumulation before the peak day of the NAAA may also cause or at least intensify the dynamical or thermodynamical anomalies, which in turn might support the formation of PM2.5 pollution over the NCP. This is a potential interesting topic that deserves further investigation in the future."

***Specific comments:***

*1. Line 88, due to emission instead of "due emission"?*

**Response:** Thank you for your careful check and help. We have revised it.

**Line 89:** "… due …" –> "… due to …"

*2. Equation (2) and Line 114-115, please explain the formula in a better way, especially for the variables. Please give descriptions for a, p, λ, and Φ? Should it be pcosΦ in Equation (2)?*

**Response:** Thank you for your careful check. We are sorry for the confusion. It should be $acos\phi$ in Equation (2). We have given descriptions for *a, p, λ,* and $\phi$ in the revised manuscript.

10    **Lines 117−118:** "… $p$ is the normalized pressure (pressure per 1000 hPa), and $a$ is Earth's radius. $\lambda$ and $\phi$ denote the longitude and latitude, respectively."

*3. Line 150-155, what are the wavenumbers 1-10, especially 5-10. Please give some description.*

**Response:** Thank you very much for your valuable advice. We have added some descriptions for wavenumbers 1−10 in

15    the revised manuscript.

**Lines 153−155:** "Wavenumbers 1−10 are the spectral components on wave-number domain produced by Fourier transform in the spatial domain. Among them, wavenumbers 1−2 represent ultra-long wave, wavenumbers 3−5 denote long wave, and wavenumbers 6−10 are synoptic waves."

20    **References**

An, X. D., Sheng, L. F., Liu, Q., Li, C., Gao, Y., and Li, J. P.: The combined effect of two westerly jet waveguides on heavy haze in the North China Plain in November and December 2015, Atmos. Chem. Phys., 20, 4667–4680, https://doi.org/10.5194/acp-20-4667-2020, 2020.

Lin, Y., Wang, Y., Pan, B., Hu, J., Guo, S., Zamora, M., Tian, P., Su, Q., Ji, Y., Zhao, J., Gomez-Hernandez, M., Hu, M., and
25    Zhang, R.: Formation, radiative forcing, and climatic effects of severe regional haze, Atmos. Chem. Phys. Discuss. [preprint], https://doi.org/10.5194/acp-2021-799, in review, 2021.

Lou, S., Yang, Y., Wang, H., Smith, S. J., Qian, Y., and Rasch, P. J.: Black carbon amplifies haze over the North China Plain by weakening the East Asian winter monsoon, Geophys. Res. Lett., 46, 452–460, https://doi.org/10.1029/2018GL080941, 2019.

30    Merryfield, W. J., Baehr, J., Batté, L. et al.: Current and Emerging Developments in Subseasonal to Decadal Prediction, Bulletin of the American Meteorological Society 101, 6, E869−E896, https://doi.org/10.1175/BAMS-D-19-0037.1, 2020.

National Research Council: Assessment of Intraseasonal to Interannual Climate Prediction and Predictability. Washington, DC:

The National Academies Press, https://doi.org/10.17226/12878, 2010.

Song, L., Wang, L., Chen, W., and Zhang, Y.: Intraseasonal Variation of the Strength of the East Asian Trough and Its Climatic Impacts in Boreal Winter, J. Climate, 29, 2557−2577, https://doi.org/10.1175/JCLI-D-14-00834.1, 2016.

Wang, J., Zhu, Z., Qi, L., Zhao, Q., He, J., and Wang, J. X. L.: Two pathways of how remote SST anomalies drive the interannual variability of autumnal haze days in the Beijing–Tianjin–Hebei region, China, Atmos. Chem. Phys., 19, 1521–1535, https://doi.org/10.5194/acp-19-1521-2019, 2019.

Yin, Z. C., Zhou, B. T., Chen, H. P., and Li, Y. Y.: Synergetic impacts of precursory climate drivers on interannual-decadal variations in haze pollution in North China: A review, Sci. Total Environ., 755, 143017. https://doi.org/10.1016/j.scitotenv.2020.143017, 2021.

Zhong, W., Yin, Z., and Wang, H.: The relationship between anticyclonic anomalies in northeastern Asia and severe haze in the Beijing–Tianjin–Hebei region, Atmos. Chem. Phys., 19, 5941–5957, https://doi.org/10.5194/acp-19-5941-2019, 2019.